# Exercise Prescription for the Work–Life Population and Beyond

**DOI:** 10.3390/jfmk8020073

**Published:** 2023-05-26

**Authors:** Gisela Sjøgaard, Karen Søgaard, Anne Faber Hansen, Anne Skov Østergaard, Sanel Teljigovic, Tina Dalager

**Affiliations:** 1Department of Sports Science and Clinical Biomechanics, Faculty of Health Sciences, University of Southern Denmark, DK-5230 Odense M, Denmark; ksogaard@health.sdu.dk (K.S.); annefaber@bib.sdu.dk (A.F.H.); or sate@pha.dk (S.T.); tdalager@health.sdu.dk (T.D.); 2Ørsted, QHSE Subject Matter Experts, DK-2820 Gentofte, Denmark; asoes@orsted.com

**Keywords:** exercise interventions, workplace health promotion, musculoskeletal disorders, musculoskeletal pain, physical capacity, strength training, resistance training, cardiovascular fitness, physical exercise training

## Abstract

The background for this paper concerns a high frequency of work-related disorders that may result from physical exposure at work being highly sedentary, repetitive–monotonous, or physically demanding. This may result in levels of physical inactivity or strenuous activity impairing health. The aim is to present an evidence-based exercise prescription for the work–life population and beyond. The exercise program is designed to be feasible for use at the workplace and/or during leisure time and to improve health, workability, productivity, sickness absence, etc. The specific concept of Intelligent Physical Exercise Training, IPET, includes the assessment of several health-related variables, including musculoskeletal disorders, physical capacity, and physical exposure at work and/or daily life activity. An algorithm with cut-points for prescribing specific exercises is provided. Exercise programs in praxis are addressed through descriptions of precise executions of various prescribed exercises and possible alternatives to optimize variation and adherence. Finally, perspectives on the significance of introducing IPET and the ongoing, as well as future lines of development, are discussed.

## 1. Introduction

Physical activity, PA, has convincingly been demonstrated to maintain health and prevent lifestyle diseases, such as cardiovascular and metabolic diseases. Leisure time PA, LTPA, has been studied extensively and proven beneficial for the prevention and treatment of musculoskeletal disorders that often may be work-related due to strenuous PA or prolonged inactivity during occupational tasks [1,2]. Large epidemiological studies show that LTPA is beneficial among workers with sedentary as well as strenuous jobs [3,4].

LTPA usually includes various sports, jogging, walking, fitness training, etc. Physical exercise training, a subset of LTPA that is planned, structured, repetitive, and performed with the intent to improve health or fitness, demonstrates the greatest benefits [5]. In sports exercise training, the programs are developed to increase capacity and function in order to optimize performance for a specific discipline, while in contrast, the physical activity at the work focuses on optimal job or task performance that may imply an adverse loading of the body [6,7]. In a health perspective it is important to perform exercises that are matched to the individual regarding daily life activity, e.g., occupational exposure–physical capacities, and health status, such as musculoskeletal disorders or other diseases. We have termed this subset of LTPA for Intelligent Physical Exercise Training, IPET [8].

The aim of IPET is to implement the knowledge from health-enhancing exercise training into work–life. IPET is novel, as such, it is optimal and composed of a 1 h exercise training program per week that is customized individually, rather than identical for all participants, as specified below. Another unique feature of IPET is that it is designed to be effective for populations in different kinds of occupations, and is currently under evaluation for populations beyond work–life due to age or health status.

Extensive literature is available for optimizing exercise training programs to increase physical capacities, such as cardiovascular fitness (CVF), muscle strength, muscle endurance, dynamic power, etc. [9]. This evidence—available from exercise physiology and sports science—is a perfect basis for planning IPET in relation to specific job demands or other daily life activities, and for designing individual training programs to attain specific outcomes by targeting, e.g., aerobic training for CVF, strength training for musculoskeletal health, and/or functional training for coordination and balance. In general, to attain such positive effects, the training intensity must be highly relative to the individual’s capacity. Implementing IPET at worksites allows sustained workability and good health, as well as prevention of sickness absence and wear and tear by increasing workers’ capacity and resilience to lifelong work exposure and beyond.

Physical exercise training increases physical capacities such as CVF, and studies since the 1980s have shown an exponential increase in the risk of mortality and death from cardiovascular diseases with decreasing CVF [10,11]. Fitness training, also termed aerobic training, positively impacts on CVF and specific health risk factors, such as BMI, body fat, waist/hip-ratio, blood pressure, blood sugar and -fat, as well as inflammatory markers (e.g., micro-CRP or CRP in blood), while strength training, also termed resistance training, additionally impacts on musculoskeletal health and balance (the latter decreasing the risk of falling). The health effects, thus, can be targeted by adjusting the combination of various training components at a sufficiently high intensity. Of note, the most recent recommendations by the World Health Organization, WHO, include moderate to high intensity aerobic and strength training [12].

Despite the well-known positive health effects of PA, a major part of the population does not meet the guidelines on moderate to high PA. According to statistics from Nordic Council of Ministers, NMR, around 60–70% of the population in the Nordic countries do not meet the recommendations [13], and similar data are available for the US and other developed countries worldwide [14,15]. A major part of the adult population spends around half of their time awake at work or commuting to work. Thus, work–life accounts for a large time fraction of that populations’ PA profile. Importantly, the PA profile must be evaluated as a collective profile across work, leisure, and sleep (Figure 1). Therefore, in general, a balance between activity and recovery is vital [6]. A major body of evidence regarding the health enhancing effects of PA has focused on LTPA and less so on occupational PA, OPA. In contrast, sedentarism at work has drawn special attention to its negative health effects, and accordingly, the focus has mainly been on physical exercise training among workers with sedentary jobs [16,17]. However, a balance between activity and recovery is likewise important among workers with heavy OPA, who may even benefit from some sedentary work tasks in combination with exercise training to strengthen their bodies to cope with the occupational exposures [18].

Musculoskeletal disorders are the most frequent cause of absenteeism in work life, early exit from the labor market, and years lived with disability [19]. Studies from the latest decades indicate that intensive strength training focused on the relevant body parts has a large potential to prevent and rehabilitate unspecific muscle and joint pain in work life [20,21].

Cascading evidence during the latest decades has resulted in the development of an algorithm for recommendation of individually tailored IPET [22,23], that will be described in detail in Section 2. The exercise training can be conducted in many different ways and, importantly, the employees must find the intervention to be “just right”, i.e., satisfy their needs and motivation for training independently, whether it is part of their work time or planned in association with their work. The allocation of a trainer or instructor to guide in planning, performing, and progressing exercises will highly facilitate the effectiveness. Additionally, instructors can give information about, e.g., exercise soreness, that is common particularly in the beginning of training, and thereby may minimize dropouts. Later, it is recommended to educate some workers to become health ambassadors to maintain motivation and ensure IPET stays an integral part of company culture. These issues will be addressed in Section 3 presenting IPET programs in praxis, and specific IPET exercises that are recommended in an Exercise Catalog in (Table A1).

In addition to these practical tools, it is important to be aware that the implementation of IPET in workplaces requests cultural changes, where time, space, and other resources are allocated for training activities for each worker to attain a lasting effect. This is independent of whether training is considered part of working hours paid by the employer or if it is provided as recommendations in relation to working time with or without payment. An important issue is that all organizational layers are involved from top management to middle managers and the workers. Attention must be paid to practical planning relative to the work tasks. Dedication from the leadership in a company to make their workers participate in IPET is decisive for the workers involvement. Middle managers have the practical role of planning training with minimal impact on time for production, including, e.g., time for a health care personal to care for patients or elderly, and still make it possible for all workers to participate in IPET for a sufficient duration. In a review paper, including Danish intervention studies on 1-h exercise training per week, productivity at baseline was found to be high and a further increase was difficult to attain [22]. Importantly, however, none of the studies identified a decrease in productivity despite the training being performed during paid working hours [22]. Notably, one of the studies detected an increase in productivity, which was reported among workers participating in more than 70% of the IPET planned [24]. In an Australian study—following the principles of IPET among computer users—similar positive effects were reported regarding productivity in terms of monetary value, as well as sickness absenteeism and presenteeism [25]. These issues are further elaborated upon in Section 4, on the perspectives of IPET.

## 2. The Concept of Intelligent Physical Exercise Training, IPET

IPET is an evidence-based conceptual model for planning individually tailored physical exercise training for people during work–life and beyond. More than 20 RCT studies published, including >4000 workers, have been conducted during the last two decades, for references see [23]. The main number of studies were performed in Denmark and listed in [6], while others are from Switzerland [26] and Australia [25,27]. Most recently, a modification of IPET for rehabilitation beyond the work–life population was developed using the same principles [28]. The IPET concept has been developed over the last two decades and a number of positive findings were evidenced, the main findings being: (1) reductions in musculoskeletal disorders; (2) reduced health risk indicators and improved physical capacities; and (3) increased productivity and monetary outcome at the company level.

IPET is based on state-of-the-art, sports science principles, and existing guidelines. The exercises are allocated to each individual based on: (1) *job profile* defined by work exposure in terms of OPA; (2) *physical capacities* in terms of CVF and muscle function; and (3) *health profile* assessed in terms of health risk indicators including musculoskeletal pain and discomfort. The theoretical background for IPET is illustrated in Figure 2, which demonstrates the interplay between job demands and the capacity of the human body. If the individual’s capacity is low, the relative workload of a specified absolute load will be high, and the internal adaptation will, in a long-term perspective, possibly result in muscular fatigue, pain, and degeneration. This will initiate the vicious circle, where the individual’s capacity is further reduced and the relative external loading for a designated work task is further increased, and vice versa; if there is a balance between work demands and the individual’s capacity, the virtuous circle starts with maintained or increased physical capacity and health. An individual’s capacity is determined and influenced by factors other than work demands, such as sex, age, anthropometry, and training status. In addition, sleep and the possibility for recovery after a workday will influence the level of capacity for the next workday.

Physical exercise training covers all types of training (aerobic training, strength training, and functional training), and from a sports science perspective, it is well documented that the different fitness areas are linked to different physiological responses (Figure 3). Thus, depending on which physical capacities or health aspects are to be improved, different types of training modes and selected exercises are to be allocated to the training program. The exercises in IPET are selected to target relevant muscles, but not to overload muscles at a risk of developing pain, e.g., due to job demands. It is known that negative effects may occur from physical exercise training in the event of an overdose; thus, dose is important both in terms of too much and too little. Therefore, it is important to highlight that IPET corresponds to not only taking the right medicine but also in the right amount for a specific disorder. No pill helps improve all disorders, but depending on the specific disorder, specified medication is prescribed. With IPET, the specified medication depends on the individual’s strengths and weaknesses (such as the cardiorespiratory system and musculoskeletal pain). Dosage of a pill is likewise important, and for IPET this corresponds to training volume in terms of intensity and duration. Additionally, recovery is important after physical exercise training. If exercising an hour per week, according to IPET, in addition to the usual activities of daily living, ADLs, there is no need for concern. However, if an individual is very active both at work and during their leisure time, it should be considered whether an additional hour of IPET is relevant, or should replace some of the high levels of LTPA.

### 2.1. The Structure of the IPET Program

To design exercise training in terms of an IPET program, the following variables are to be identified:Individual job profile: categorized into sedentary, standing/walking, or physically heavy.Physical capacity profile: aerobic capacity, muscle strength, and balance.Health profileBody composition: BMI, or waist/hip ratio. If more sophisticated measurements are available, body fat percentage, blood pressure and blood profile may offer additional and more precise estimates.Musculoskeletal pain: localization and pain intensity in one or more of six body regions (hands/elbow, neck, shoulder/upper back, lower back, hip, knee/ankle).

Sedentary work, for example, includes most types of office work (with or without screen work), laboratory technicians and chauffeurs. Standing/walking work includes cleaners, health care workers without lifting/carrying tasks, nurses, gardeners, and renovation workers, as examples. Job groups within construction, manufacturing, and public safety (e.g., firefighters) typically identify their OPA as physically heavy. Based on the exposure (job profile) and individual profile (physical capacity and health assessment), the amount of aerobic training, strength training, and functional training are determined. The presence of musculoskeletal pain or low muscle strength determine which specific strength training exercises should be included in the training program depending on the body part of concern. Figure 4 shows an overview of the IPET concept. Overall, IPET is composed of 1 h training a week composed of a 10 min warm-up, 20 min specific job profile training, and 30 min specific individual profile training.

### 2.2. The IPET Algorithm

The individual exercise training program is designed based on categorization of job profile, and assessments of both physical capacity and various health parameters, including information on musculoskeletal pain. Objective assessments that are performed as an individual health check have been utilized in IPET research projects; however, simpler methods may suffice in practice. Thus, Table 1 indicates which measures of self-reports and self-assessments, or more advanced objective assessments, will result in the allocation of the various types of training. The specific cut-point recommendations for questionnaire replies, self-assessments, and/or objective measures are based on national and international recommendations and presented in Table 2 These are mainly derived from Danish population studies [8,29] and cut-points from other populations are to be included whenever available.

Once information on the job profile and individual profile—in terms of a combination of physical capacity—and the health profile have been assessed, the allocation of specific types of training can be deduced. In particular, the IPET algorithm determines the training mode and duration for the “individual profile” (Figure 4 and Table 3). Training exercises for warm-up, job profile, and specific strength- and functional-training can be seen in Appendix A (Table A1). The training, according to the individual profile, is determined by the cut-points that are exceeded for physical capacity and health profiles. Several variables may result in the allocation of a training category, e.g., BMI and blood sugar for aerobic training, or low arm/shoulder strength and shoulder pain for shoulder strength training, see Table 1; however, such variables often correlate and the number of cut-points recommending a training category are therefore disregarded.

The IPET algorithm operates in 5 min periods, allowing for up to six different aerobic, strength, or functional training exercises during the 30 min of individual profile training. The exercises in Appendix A (Table A1) are listed in a prioritized order, implying that the first exercise on the list should be performed first. However, if an exercise aggravates discomfort or some of the other suggested exercises target the experienced symptoms better, another exercise on the list may be chosen. For strength training, the algorithm maximally allows exercises prescribed for three specific body regions at the same time. Furthermore, if the individual is recommended an all-round strength training, then only two body parts are prioritized for specific body part strength training. If pain above the cut-point is reported in more than three body regions, the pain intensity defines the selection of the two most painful body regions. When the prescribed exercises have been performed for a while, they may be exchanged with other exercises for the same muscle group.

It is important to point out that suspicion of serious illness or dysfunction for which IPET will not be beneficial, will result in a consultation with a general practitioner, chiropractor, or physiotherapist, depending on the suspicion, and that reservations are possible for minor adjustments.

## 3. Exercise Programs in Praxis

When starting IPET, it is recommended for an instructor to provide live instructions via video, an App, etc., to show exercises and ensure appropriate and confident self-administered training [31]. Strength training may be unexplored territory for many; therefore, strength training may require some adaptation to reach the desired volume and intensity. In older adults, strength training may be introduced particularly intensively, preferably using supervised sessions at least two times a week. On the contrary, aerobic activities are often more simple, well-known ADLs, and therefore require less instruction or learning of new skills. If available, an instructor may at regular intervals supervise workouts to maintain motivation and ensure sufficient exercise intensity.

While IPET has primarily been researched across the work–life population, the principles and approaches are admissible in older adults (>65 years). As there are differences between the working and the elderly populations, e.g., OPA versus ADL, an altered version of IPET has been developed and is currently being tested in a three-arm multicenter, randomized controlled trial among older adults [28]. It is generally recommended to consult with a general practitioner, chiropractor, or physiotherapist, if there is a suspicion of serious illness or dysfunction where IPET will not be beneficial. In particular, medical attention should be sought prior to engaging in IPET if the following symptoms are present: (1) Shortness of breath at rest or/and pain or heaviness in the chest/left arm; (2) Blood pressure: systolic >180 or diastolic >110 mmHg; (3) Fever, swelling or redness in any joint with no history of injury (or multiple inflamed joints); (4) Changes in vision (blurriness or loss of sight); (5) Sudden weakness; (6) Unexplained weight loss or continuous fatigue; or (7) Neurological symptoms.

### 3.1. Exercise Progression

Once exercise training has successfully been initiated, it is important to adjust the specific prescription continuously, but slowly, to improve physical capacities. The relative intensity may stay the same throughout long periods, but the absolute intensity (e.g., speed, load, complexity) will increase over time, in order for the cardiorespiratory, musculoskeletal, and neural systems to improve.

Strength training progression is guided by the number of repetitions and intensity, in terms of the percentage of one repetition maximum (Table 4). An adaptation phase over the first four weeks is recommended for novice trainees but may be omitted for individuals who already engage in regular strength training activities. After the initial first month it is important that training sessions still progress in the number of sets and intensity. As a rule of thumb, the load should be increased when more than two additional repetitions to the prescribed number can be performed. However, the progression listed in Table 4 does not apply for static strength training exercises, such as the plank, where it is the duration or technique that determines the exercise intensity. In fact, the starting position and technique can alter the intensity of both dynamic and static strength training exercises. The plank, for example, may be performed on the knees/forearms, toes/forearms, or toes/hands, offering different options for progression and regression.

Aerobic training intensity and exercise progression can be guided by heart rate measurements or by rating the perceived exertion, RPE, using different RPE scales. As illustrated in Figure 5, RPE may be assessed using the original 6–20 point Borg scale (C) or the reconstructed 0–10 point version of the scale (C-R) [32,33]. To improve aerobic capacity, the intensity should generally be strenuous, corresponding to above 70% of heart rate reserve, HRR, or 14–15 (C)/7–8 (C-R) on the Borg scale for continuous, steady-state aerobic exercise activities. The required exercise duration may be optimized even further by increasing the intensity to above 85% of HRR, or 17(C)/9(C-R) on the Borg scale. At these high intensities, activity needs to be intermittent, as it can only be sustained for very short periods of time without rest breaks. High intensity interval training (HIIT) may be a very time effective choice [34].

The general principles for the various parts of the IPET exercise program are listed in Table 5: (1) Warm-up; (2) training according to job profile/OPA or ADL level; and (3) individual profile specific aerobic, strength and functional training. The specific exercises are listed and explained in the (Table A1).

### 3.2. Exercise Participation

An important aspect is the association between training participation, also termed training adherence or attendance, and the positive effects of IPET. Feasibility studies may be performed for specific work settings to reveal optimal implementation strategies to attain high acceptability, sustainability, adherence, and compliance [35]. Previously, a training adherence threshold of 70% has been recommended to attain clinically meaningful benefits and is often utilized as a pre-set value for per-protocol analysis in studies [24,29,36,37]. To attain a threshold of 70% adherence, it is important to implement initiatives before and during an intervention with IPET to maintain engagement and motivation. Further, exercise compliance, i.e., the correct performance of the exercises, has been shown to be vital [27].

Research has shown that the same effect is achieved, regardless of whether all training is performed once a week or divided into shorter periods performed over more sessions, as long as the same total training volume is achieved with the right intensity [38,39,40]. This offers major flexibility in planning the training, which may facilitate participation at the workplace. In addition to IPET, it is recommended that each individual meets the international recommendations from the WHO on physical activity (at least 150 min per week with moderate intensity, or at least 75 min per week with high intensity, or a combination of both). IPET can be regarded as being equivalent to high intensity.

## 4. Conclusions and Future Directions

IPET has developed in a dynamic process of compelling results from a large number of randomized controlled studies among populations with different job profiles [6,22,23]. While the basic principles in the concept have been fundamental, the actual design and implementation of the training has been adapted for each of the target population, always meeting their workplace conditions and facilities. This comprehensive explorative process has shaped the IPET concept into its current state, where it is ready to be upscaled and offered to the working population in general and beyond. The concept can be developed and offered as a digital solution to create individually designed training programs based on the algorithm described above regarding job exposure, physical capacities, and health variables.

While there is consistent evidence of the positive effects for the adherent individuals across job sectors, we require further knowledge on three important aspects for a successful implementation: (1) shaping IPET to smoothly fit into daily life and work culture in each workplace; (2) including motivational features for the individual worker; and (3) evidence of the economic benefit to motivate the company.

### 4.1. Fitting IPET into Workday and Culture

An intelligent feature of IPET is the restriction to only 1 h weekly planned exercise training at the workplace either during work time, or immediately before or after. For workers with physically heavy work tasks, it may be convenient to schedule the training in the beginning of workdays, since it may serve as a substantial warm-up before starting work. Beneficial effects have been shown for training performed as one whole hour or split into periods of 6 × 10 min, 5 × 12 min, 3 × 20 min [38]. Training can also be performed with or without an instructor, with similar effects, as long as the initial instructions are carefully conducted [38]. IPET may happen at the workplace, the actual workstation, in company fitness facilities, in nearby fitness centers, or simply, planned training anywhere with colleagues that is workplace initiated. However, one of the most recent studies clearly showed a much higher adherence when IPET was performed at the workplace with colleagues, compared to alone at home [35]. This offers flexible ways to fit IPET into the workday routines, and Danish intervention studies with IPET performed during working hours show that this does not compromise work productivity [24,41].

The workplace can contribute through campaigns and events to initiate and motivate workers to perform training at the workplace or at leisure. Employers can also support the use of IPET at leisure, e.g., by full or partial reembursement of fitness or sports club memberships on the condition of regular use. The payments can even be in terms of days off for special physical training performances; the rationale being for the employer that a fit worker has less absenteeism.

One of the important barriers for participation in IPET at the workplace during worktime is a lack of time and difficulty in prioritizing IPET relative to other responsibilities [42]. The most important factor for overcoming this barrier is support from the closest leader and the acceptance, and even appreciation, of taking the time to maintain workability [43]. Overcoming the issue of prioritizing may be done by integrating IPET into the daily routines to the largest possible degree, so that it is not a conscious choice of every session but rather just one of the tasks in a normal workday routine.

### 4.2. Motivation of the Individual Worker

The effect of IPET highly depends on the individual’s motivation for participation. Employers can contribute to motivating their employees to adhere to IPET by providing time and space to the IPET initiative. Additionally, the appointment of dedicated health ambassadors at the workplace can provide important motivation and further enhance the support from a strong collegial network [44].

Changes in lifestyle may be supported with cognitive behavioral training [45]. This has been developed to support an increase in activity regarding treatment and interventions among chronic neck pain patients [46,47]. Recently, a digital version was developed in an mHealth solution for low back pain patients to help them stay active in spite of pain [48,49]. In former studies it has also been used to support IPET [50,51], and in future digital versions of IPET, cognitive behavioral support may be further adapted based on the individual worker’s need for support, depending on their readiness to change [52].

Maybe, most importantly, an essential part of the individual’s motivation is for the implementation to be a participatory process. The exercise physiological basics in IPET are mandatory, but there is a large flexibility regarding how to reach these basics. For instance, heart rate can be raised by many different activities, dancing, as well as skipping and running, depending on personal preferences. Additionally, a larger catalog of alternative types of exercises may be incorporated, offering a possibility for variety, as well as personal choice.

### 4.3. Economy from a Company Perspective

To motivate the company and employers to make the effort to implement IPET, and support their employees in participating, there is a need for documenting the effect on productivity, presenteeism and absenteeism, as well as other positive effects that provide profit on the bottom line. Furthermore, evidence of a long-term effect is essential. Only a few studies have followed IPET for more than 12 months, but in a company-driven experiment, positive effects proceeded over a 3 year period [53]. As earlier mentioned, in the IPET interventions when productivity was measured, it was maintained, and in some studies even increased, in spite of training being performed during working time [54]. One Danish IPET study found for office workers with a high adherence at a minimum of 70%, that they had a reduction in both productivity loss at work and sickness absence [24]. In an Australian IPET study on office workers, evidence of possible productivity benefits were found in both a general population of office workers, as well as among those with neck pain. The study concludes that a combination of workplace ergonomics and neck-specific exercise training for office workers is a sound financial investment and business strategy with longer-term gains [25]. Similarly, a Swiss study found an effect of an IPET inspired intervention on neck pain-related work productivity, as well as a weekly saved cost of CHF 27.40 per participant, and concludes that the intervention could reduce the economic burden of neck pain among office workers [26]. However, regarding economic evaluation of *workplace physical activity interventions* in general, two systematic reviews recently pointed out that such evidence is inconsistent, due to large heterogeneity in such studies [55,56].

### 4.4. Section Summary

More than 20 RCT studies have been leading to the IPET version described in the present paper, showing compelling evidence of numerous positive effects across many job sectors. However, as the IPET concept is new and has not yet been widely used, the evidence is still limited. The present detailed outline of how to use the IPET concept will hopefully inspire further research, allowing for the systematic reviews and meta-analysis requested to present the highest level of evidence for the effect.

Furthermore, future studies are needed to reveal if population specific cut-points can optimize the IPET algorithm, if some ways of fitting IPET into daily life will work better than others, and if specific recommendations may become relevant for different types of jobs and organizations. Employees should be involved in the further development and adaption of IPET towards their specific needs and wishes for health promotion, so that exercise training is not seen as a curse, and instead as the cure it is intended to be. The flexibility in the program has already been used to target other areas than the work–life population, i.e., the elderly in rehabilitation. This can be extended to offer relevant, evidence-based, individual training programs to other patient groups, such as those with diabetes, cardiorespiratory disorders, or obesity. Finally, evidence on the productivity and economic benefits of IPET interventions is positive but scarce. To convince employers and policy makers, more studies on the economic benefits of IPET, including both cost-benefit and cost-utility analysis, need to be conducted.

## Figures and Tables

**Figure 1 jfmk-08-00073-f001:**
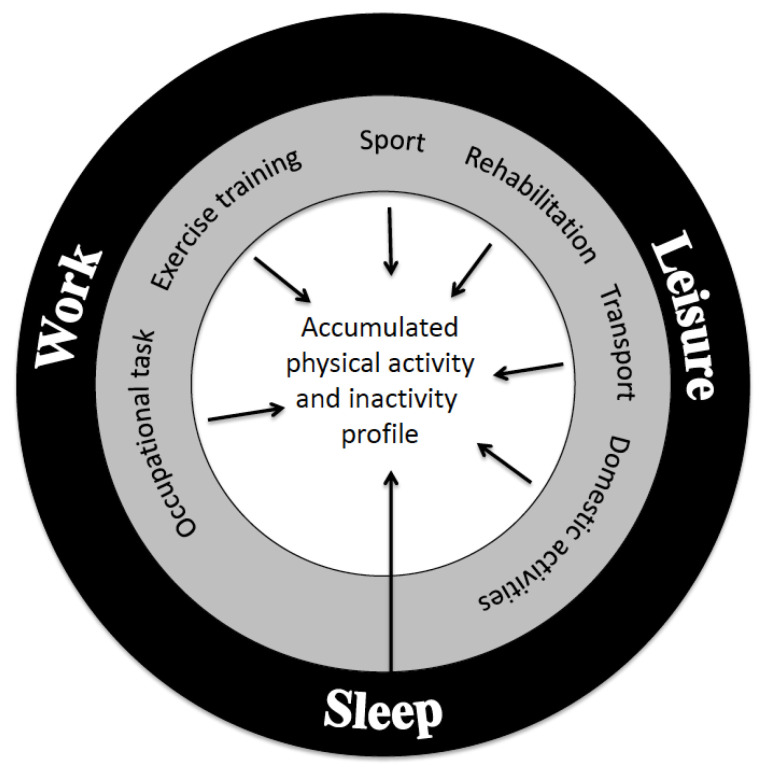
The daily profile of physical activity (PA) consists of three domains of work, leisure time, and sleep alongside with their subdivisions, each with its positive or negative contributions to health and physical capacity [6]. Reproduced with permission from Exercise Sport Sci Rev.

**Figure 2 jfmk-08-00073-f002:**
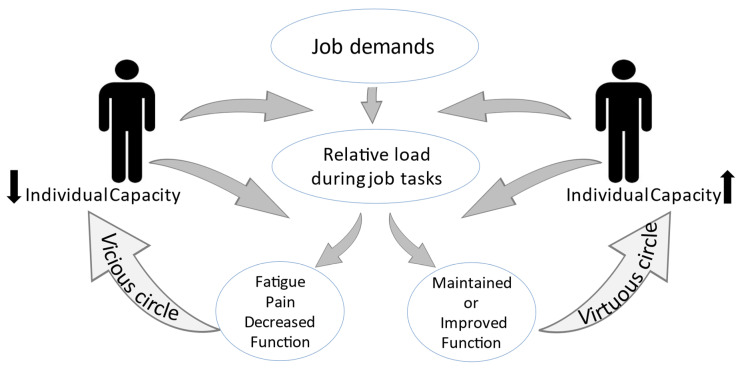
Vicious and virtuous circle model presents the importance of the individual’s capacity for the relation between a designated work task and its load imposed on the body. The relative load of the workdays may, in a longer term, cause either an increase or decrease in individual capacity, thereby initiating a vicious or a virtuous circle. Adapted from [7].

**Figure 3 jfmk-08-00073-f003:**
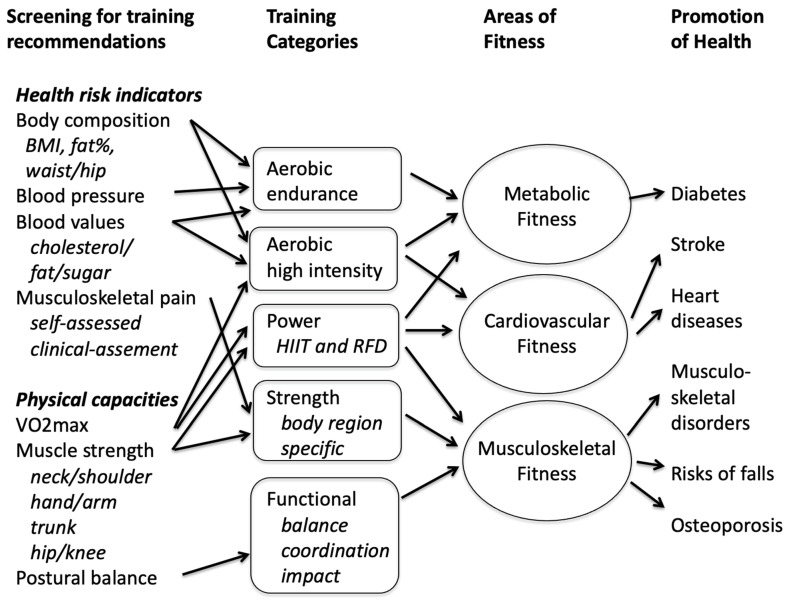
Overview of screening recommendations, training categories, and their effects. HIIT: High intensity interval training. RFD: Rate of Force Development. Adapted from [8].

**Figure 4 jfmk-08-00073-f004:**
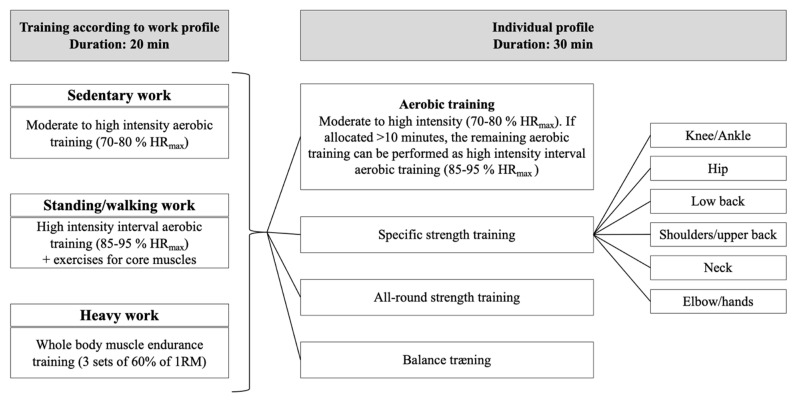
Cartoon of the conceptual model of IPET. The IPET concept is based on 50 min effective intelligent exercise training each week, excluding warm-up time for around 10 min. Twenty minutes is allocated to the individual’s work profile, and 30 min is allocated to target the individual’s physical capacity and health profile. In addition to the 50 min IPET, warm-up exercises are recommended to increase the body’s temperature and to start the circulation, so that the blood supply increases to the working muscles. The intensity is light, up to 60% of maximum heart rate, corresponding to 9–13 on the 6–20 Borg scale.

**Figure 5 jfmk-08-00073-f005:**
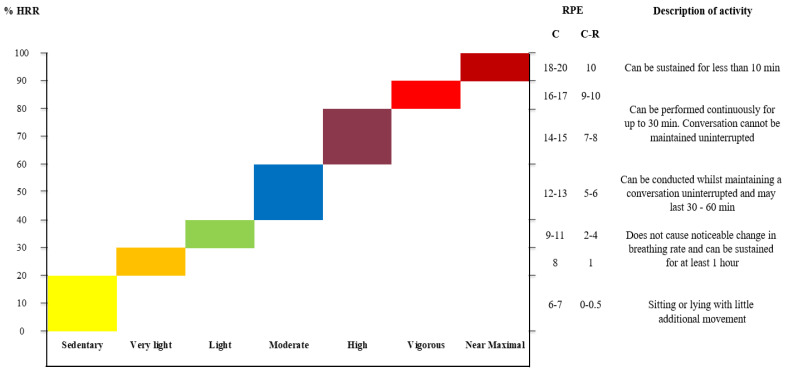
Overview of aerobic training intensities measured objectively as heart rate reserve capacity (%HRR) and/or using the corresponding subjective perception. The rate of perceived exertion (RPE) can be evaluated from the 6–20-point Borg scale (C) or the reconstructed 0–10-point Borg scale (C-R). Adapted from [7].

**Table 1 jfmk-08-00073-t001:** Overview of measures leading to prescription of various exercise categories.

		Measurement Methods		Aerobic Training	Strength Training							Functional Training
		*Self-Report or Self-Assessed Methods*	*Advanced Methods*		Elbow/Hand	Neck	Shoulder/Upper Back	Lower Back	Hip	Knee/Ankle	All-Round	
**Job profile**	Sedentary work	Questionnaire	Observation	X								
	Standing/walking work	Questionnaire	Observation	X								X
	Physical hard work	Questionnaire	Observation								X	
**Physical capacity profile**	Cardiovascular fitness	Aerobic capacity		X								
			Submaximal VO_2_max test	X								
			Maximal VO_2_max test	X								
	Muscle strength	General strength									X	
			Neck extension			X						
			Shoulder elevation				X					
			Arm abduction				X					
			Back flexion/extension					X				
			Leg flexion/extension						X			
			Hand grip		X							
	Balance and function	Balance test	Balance test									X
**Health profile**	Health indicators	Body mass index, BMI	Body mass index, BMI	X								
			Fat percentage	X								
		Hip/waist ratio	Hip/waist ratio	X								
			Blood pressure	X								
			Blood sugar	X								
			Blood cholesterol	X								
	Musculoskeletal disorders	Elbow/Hands			X							
		Neck				X						
		Shoulder/Upper back					X					
		Lower Back						X				
		Hip							X			
		Knee/Ankle								X		

**Table 2 jfmk-08-00073-t002:** Cut-points leading to prescription of IPET.

	Variable	Measurement	Description of Method	Cut-Point/Group Allocation	
**Job profile**	Occupational exposure category	Self-report	How would you categorize the physical demands in your main job? (1) *Mainly sedentary work that does not require physical exertion* (2) *Work largely performed while standing or walking* (3) *Standing or walking work with some lifting and/or carrying activities* (4) *Heavy or fast-paced work, which is strenuous*	Sedentary Standing/walking Physically heavy Physically heavy	
		Observation	Occupational exposure may be assessed with real-time observation/video recording, electromyographic activity, activity types using accelerometery or cardiorespiratory exposure using heart rate monitoring	Should be decided based on the specific job group	
**Physical capacity profile**	Cardiovascular fitness	Self-report	*How would you score your aerobic capacity compared to others of your own age and sex?*	Responses below 5 (corresponding to average) lead to prescription of extra aerobic training	
			Assessed on an 0–10 NRS ranging from as poor to as good as possible		
		Objective assessment	Direct or indirect assessment of relative VO_2_max (mL O_2_/kg/min). The test may be performed as a maximal or submaximal assessment	“Low” capacity leads to prescription of extra aerobic training	
				Women (years)	Men (years)
				20–29 ≤ 34	20–29 ≤ 43
				30–39 ≤ 33	30–39 ≤ 39
				40–49 ≤ 31	40–49 ≤ 35
				50−59 ≤ 28	50–59 ≤ 31
				60–69 ≤ 26	60–69 ≤ 26
				≥70 ≤ 26	≥70 ≤ 24
	Muscle strength	Self-report	*How would you score your strength capacity compared to others of your own age and sex?*	Responses below 5 (corresponding to average) lead to prescription of extra all-round strength training	
			Assessed on a 0–10 NRS ranging from as weak to as strong as possible		
		Objective assessment	Objective strength assessments for specific body types.	Additional strength training is prescribed if an individual scores below 80% of the average population	
			Strength tests previously used in IPET trials include neck extension, shoulder elevation, arm abduction, back flexion/extension, leg flexion/extension and hand grip		
	Balance and function	Self-assessment	Self-assessed standing time on non-dominant leg with the arms crossed over the chest (hands on the shoulders)	If the individual loses balance before reaching 30 s, functional training is prescribed. Three trials are given	
**Health profile**	Body mass index	Self-assessed or self-report	Assessed as an individual’s weight in kilograms divided by the square of height in meters	A body mass index above 25 leads to prescription of extra aerobic training	
	Waist/hip ratio	Self-assessed	Waist circumference is the distance around the waist from between the lowest rib and the upper iliac crest. Hip circumference is the widest distance of the buttocks. The ratio is calculated by dividing the waist measurement with the hip measurement.	Ratios above 1 for men and 0.8 for women lead to prescription of extra aerobic training	
	Fat percentage (%)	Objective assessment	May be assessed using a bioimpedance scale, skinfold measurements, Dual-Energy X-ray Absorptiometry (DXA) and hydrostatic weighing (offering various accuracy)	Individuals who score “high” get prescription of extra aerobic training	
				Women (years)	Men (years)
				20–39 ≥ 33%	20–39 ≥ 20%
				40–59 ≥ 34%	40–59 ≥ 22%
				50–99 ≥ 36%	60–79 ≥ 25%
	Blood pressure (mmHg)	Objective assessment	Blood pressure is assessed using a blood pressure monitor	Individuals with readings above 140 mmHg (systolic) and/or 90 mmHg (diastolic) get additional aerobic exercise prescribed	
	Total blood cholesterol	Objective assessment	Cholesterol is assessed by obtaining a blood sample (fasted state)	Total cholesterol > 5.0 mmol/l leads to prescription of aerobic training	
	Blood sugar (glucose)	Objective assessment	Blood glucose is assessed by obtaining a blood sample (fasted state)	Blood glucose levels below 4 and above 7 mmol/l leads to prescription of extra aerobic training	
	Musculoskeletal disorders	Self-report	*For how many days in total have you had (specific body part) pain during the last 3 months/7 days? **	Any symptoms lead to prescription specific strength training for the body part.	
			*If symptoms have been prevalent for one or more days, a follow up question is asked about average pain intensity: Please indicate your level of (specific body part) pain within the last 3 months/7 days*	If an individual has symptoms in more than two (tree) body parts, the pain intensity guides the prioritization.	

The described cut-points are mostly derived from the Danish population [8,22,30] and may be slightly different in other countries that are to be included whenever available. * The recall period for musculoskeletal symptoms may vary depending on the specific program and duration of the intervention. Previous IPET trials have mostly used 3-month or 7-day recall periods. NRS: Numerical rating scale. The phrasing of the questionnaires is marked in italics. The health profile is composed of a musculoskeletal disorder and a number of health risk indicators, see also Table 1.

**Table 3 jfmk-08-00073-t003:** Algorithm for allocation of IPET individual profile.

Number of Cut-Points Exceeding the Recommended	Combinations	Aerobic Training	Strength Training			Functional Training
			Specific 1	Specific 2	All-Round	
		AT	S1	S2	Sa	FT
0		15			15	
1	AT	20			10	
	S1		20		10	
	S2			20	10	
	Sa	10			20	
	FT				20	10
2	AT+S1	15	15			
	AT+S2	15		15		
	AT+Sa	15			15	
	AT+FT	15			5	10
	S1+S2		15	15		
	S1+Sa		15		15	
	S1+FT		15		5	10
	S2+Sa			15	15	
	S2+FT			15	5	10
	Sa+FT				20	10
3	AT+S1+S2	10	10	10		
	AT+S1+Sa	10	10		10	
	AT+S1+FT	10	10			10
	AT+S2+Sa	10		10	10	
	AT+S2+FT	10		10		10
	AT+Sa+FT	10			10	10
	S1+S2+Sa		10	10	10	
	S1+S2+FT		10	10		10
	S1+Sa+FT		10		10	10
	S2+Sa+FT			10	10	10
4	AT+S1+S2+Sa	10	10	5	5	
	AT+S1+S2+FT	10	10	5		5
	AT+S1+Sa+FT	10	10		5	5
	AT+S2+Sa+FT	10		10	5	5
	S1+S2+Sa+FT		10	5	10	5
5	AT+S1+S2+Sa+FT	10	5	5	5	5

AT = Aerobic training. S1 = specific strength training, first priority. S2 = specific strength training, second priority. Sa = all-round strength training. FT = Functional training. Numbers showcase minutes, i.e., 5–20 min of a specific training type. NB: if a person exceeds all cut-points and reports pain in all body regions, the two most painful body regions are prioritized (Exercises for neck, shoulder/upper back will be collapsed, as well as low back and hip).

**Table 4 jfmk-08-00073-t004:** Strength training progression.

Week	Number of Sets per Exercise	Repetition Maximum, % (Number of Repetitions)
**1**	2	60% (15)
**2**	2	65% (15)
**3**	3	65% (15)
**4**	2	70% (12)
**5**	3	70% (12)
**6**	4	70% (12)
**7**	2	75% (10)
**8**	3	75% (10)
**9**	4	75% (10)
**10**	2	80% (8)
**11**	3	80% (8)
**12**	4	80% (8)
**13**	4	80% (8)
**14**	2	80% (8)
**15**	3	80% (8)
**16**	4	80% (8)

Overview of the development in intensity and repetitions in the initial 16 weeks of doing IPET. The intensity is indicated by the percentage of 1 repetition max (RM), which is the load that can maximally be lifted for one repetition. As the intensity increases to a higher percentage, the number of repetitions naturally needs to decrease as indicated in parentheses.

**Table 5 jfmk-08-00073-t005:** Overview of intensity and progression principles for the different parts of IPET.

Part of Program	Principles
**Warm-up**	Warm-up intensity is maximally 60% of HRR, or 9–13 on the C (6–20) scale, meaning that activities can be conducted whilst maintaining a conversation
**Job profile/** **activity of daily living**	**Sedentary:** Aerobic exercise activities may be chosen randomly, but should be performed for a minimum of 10 min each Exercise intensity is 70–80% of HRR, or 14–15 on the C (6–20) scale, meaning that conversation cannot be maintained uninterrupted The activity should be a performed steady state (continuously for the full 20 min duration without rest breaks)
	**Standing/walking:** Exercise intensity is 85–95% of HRR, or 17–19 on the C (6–20) scale, meaning that exercise that cannot be sustained for more than a few minutes without rest The activity should be performed as intervals that may follow the below work/rest ratios (1:2): - Active 30 s, recovery 60 s - Active 20 s, recovery 40 s
	**Heavy:** Exercises should be performed for three sets of 15–20 repetitions with an intensity corresponding to 60% of 1 RM The order of exercises can be switched as preferred, and it is recommended to alternate between exercises (performing super sets or giant sets (circuits))
**Aerobic training**	Performed as continuous AND/OR interval-based training depending on personal preferences. **Continuous:** Exercise intensity is 70–80% of HRR, or 14–15 on the C (6–20) scale, meaning that conversation cannot be maintained uninterrupted Should be performed as continuous activity for the full duration of the prescribed aerobic exercise **Interval-based:** Exercise intensity is 85–95% of HRmax, or 17–19 on the C (6–20) scale, meaning that exercise that cannot be sustained for more than a few minutes without rest The activity should be performed as intervals that may follow the below work/rest ratios (1:2): - Active 30 s, recovery 60 s - Active 20 s, recovery 40 s
**Specific strength training**	The exercises in Table A1 are listed in a prioritized order, but may be exchanged based on preferences and local discomfort Exercise intensity is 65–80% of 1 RM. When the prescribed repetitions can easily be performed, the load and/or the difficulty of the exercise should be increased Most exercises can be scaled in intensity by using elastic bands with more/less resistance (different colors) The order of exercises can be switched as preferred, and it is recommended to alternate between exercises (performing super sets or giant sets (circuits))
**Functional training**	The exercises are listed in prioritized order but may be exchanged based on preferences Each exercise should take ~5 min. Perform the exercise 3–4 times for 30 s with 30 s rest between each attempt. Always scale the exercise so it can be completed with effort

RM: Repetition maximum. HRR: Heart rate reserve capacity. The C (6–20) scale refers to the Borg-scale of perceived exertion which is elaborated in Figure 5.

## Data Availability

Not applicable.

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
