# Peer review of "Exercise Prescription for the Work–Life Population and Beyond"

_jfmk, 2023, doi:10.3390/jfmk8020073_

Round 1
Reviewer 1 Report
This review is interesting as it provides a consolidated information on physical activity for the work-life population.
The title is catchy but not much focus on "beyond", hence its appropriate the words "and beyond" be removed.
Should include a table of content.
I find the information is put-forth in few easy to follow subheadings, which is commendable, however, the scope of the review is not well written. Perhaps adding an introductory on the rational and aims of the review which eventually lead to the issues discussed within each subheading would ease reading and improve flow.
I noticed the font is inconsistent, see line 64 and few other places. Please check.
I also suggest the manuscript is proof read by a native English speaker.
I also suggest the manuscript is proof read by a native English speaker.
Author Response
Reviewer comments:
This review is interesting as it provides a consolidated information on physical activity for the work-life population.
The title is catchy but not much focus on "beyond", hence its appropriate the words "and beyond" be removed.
Author reply: It is of importance for us to emphasize that the concept presented is relevant not only for the work-life population, but also for those beyond work-life, e.g. healthy ageing or rehabilitating. As the other referees did not suggest any changes in the title, we prefer to keep the title in its present form.
Should include a table of content.
Author reply: A table of content has been added
I find the information is put-forth in few easy to follow subheadings, which is commendable, however, the scope of the review is not well written. Perhaps adding an introductory on the rational and aims of the review which eventually lead to the issues discussed within each subheading would ease reading and improve flow.
Author reply: The introduction (section1) is an introductory as requested and we have now more explicitly added the aim of the paper ahead of the following paragraphs detailing the issues discussed within each subheading (section 2-4). We hope this now will ease the reading.
I noticed the font is inconsistent, see line 64 and few other places. Please check.
Author reply: We have checked the font throughout to be consistent.
I also suggest the manuscript is proof read by a native English speaker.
Author reply: A native English speaker has proof read the manuscript and we have revised accordingly.
Reviewer 2 Report
General recommendations
This review has failed to demonstrate the effectiveness of the IPET program as the authors have not used an evidence-based methodology for that purpose.
Specific recommendations
Abstract
The authors do not indicate the following data:
They do not indicate any background justifying the interest in dealing with the analyzed subject matter, nor whether the term IPET had been previously described in other scientific works.
What type of article they have written: a clinical trial, narrative review, systematic review or others.
Development of the entire document
Some paragraphs are found with typeface and/or boldface, this should be standardized.
The meaning of LTPA or OPA has not been found, authors are reminded that this is a practice that is not recommended in an article that should be peer-reviewed.
It is not recommended to talk about the term IPET without having presented its definition, therefore, the content of this section should be reorganized.
In this section the authors do not identify what is new about an article on IPET.
I understand that everything necessary to eliminate delayed onset muscle soreness is done (Hotfiel T, Freiwald J, Hoppe MW, Lutter C, Forst R, Grim C, Bloch W, Hüttel M, Heiss R. Advances in Delayed-Onset Muscle Soreness (DOMS): Part I: Pathogenesis and Diagnostics. Sportverletz Sportschaden. 2018;32(4):243-250, among others).
This section does not differentiate between problems caused by incorrect exercise performance vs. high training load. This information should be clarified.
In section 2.1, the IPET program is mentioned, where are the characteristics of this concept defined?
Figure 2.3 indicates the conceptual model of IPET, however, it does not indicate whether this model has been validated and what its objectives are in terms of the population that will be used.
Table 2.2 should be oriented horizontally and not vertically. If the authors look at this table it cannot be seen correctly.
The authors should indicate how they have table 2.2 in more detail.
Authors should submit this topic and use the protocol of a systematic review to achieve an acceptable level of evidence. The following link provides information on this subject: http://www.equator-network.org/
Performing an analysis of a topic without performing a search protocol of the selected articles makes the review have a biased view of the subject matter, this is because the authors' standard of the MDPI journals always indicates that they should receive systematic reviews, not bibliographic or narrative reviews.
Authors can also use the AMSTAR 2 protocol (Shea BJ, Reeves BC, Wells G, Thuku M, Hamel C, Moran J, et al. AMSTAR 2: a critical appraisal tool for systematic reviews that include randomised or non-randomised studies of healthcare interventions, or both. BMJ. 2017;358:j4008) if they consider PRISMA to be complex.
That said, authors should adapt all sections of this review or any of the suggested methodologies. To achieve that goal, this reviewer advises the authors to take a more specific and not so general approach. Perhaps it would be more interesting to focus on a specific type of population (e.g., noncommunicable diseases), as this would make it easier to demonstrate the efficacy of this program.
In the event that the authors do not agree to adapt this review to the PRISMA or AMSTAR 2 methodology, they can present to this reviewer according to the checklist advised by these guidelines how many items this review complies with and make the necessary changes so that this is reflected in the document. However, authors should be aware that this aspect should be included in a limitations section.
The section on conclusions and future directions is usually done separately, not as a whole. However, according to the information that this review has been able to demonstrate, this section is too long and too broad.
References
Thirty percent of the literature is more than 10 years old and within this percentage are examples prior to the year 2000.
There are exceptions in which it is necessary to use older bibliography, however, this must be justified by the authors, since it is assumed that the articles presented should be those of greater novelty and relevance found within the subject under study.
Author Response
Comments and Suggestions for Authors
General recommendations
This review has failed to demonstrate the effectiveness of the IPET program as the authors have not used an evidence-based methodology for that purpose.
Author reply: The present paper is a state of knowledge with specific emphasis on a unique practical exercise prescription for the work-life population and beyond. The journal formalities did not allow to include a more established evidence-based methodology type of review for the demonstrated effectiveness of IPET. Based on this referee’s request we have now in the “section 2. The Concept of Intelligent Physical Exercise” added a paragraph including the most essential references e.g. also two review papers on IPET.
Specific recommendations
Abstract
The authors do not indicate the following data:
They do not indicate any background justifying the interest in dealing with the analyzed subject matter, nor whether the term IPET had been previously described in other scientific works.
Author reply: We have now added a sentence as background justifying the topic in this manuscript. IPET has been described in numerous papers earlier and a reference is given on the first published version of this concept from 2014 already in the second paragraph of the introduction. However, as it is unusual to present references in an abstract, we do not add a reference here.
What type of article they have written: a clinical trial, narrative review, systematic review or others.
Author reply: The present paper is a state of knowledge with specific emphasis on a unique practical exercise prescription for the work-life population and beyond.
Development of the entire document
Some paragraphs are found with typeface and/or boldface, this should be standardized.
Author reply: We have checked the font throughout to be consistent.
The meaning of LTPA or OPA has not been found, authors are reminded that this is a practice that is not recommended in an article that should be peer-reviewed.
Author reply: LTPA and OPA were defined abbreviations in the introduction and they are well established standard abbreviations commonly used in this area of research.
It is not recommended to talk about the term IPET without having presented its definition, therefore, the content of this section should be reorganized.
Author reply: IPET has been described in numerous papers earlier and a reference is given on the first published version of this concept from 2014 already in the second paragraph of the introduction.
In this section the authors do not identify what is new about an article on IPET.
Author reply: In the introduction the aim has now been specified and the novelty has been emphasized. For instance, a unique feature of IPET is that it is designed to be applied to work life populations across different kind of occupations, pain sites and currently under evaluation also for populations beyond working life age.
I understand that everything necessary to eliminate delayed onset muscle soreness is done (Hotfiel T, Freiwald J, Hoppe MW, Lutter C, Forst R, Grim C, Bloch W, Hüttel M, Heiss R. Advances in Delayed-Onset Muscle Soreness (DOMS): Part I: Pathogenesis and Diagnostics. Sportverletz Sportschaden. 2018;32(4):243-250, among others).
This section does not differentiate between problems caused by incorrect exercise performance vs. high training load. This information should be clarified.
Author reply: We agree that soreness and/or injury are important to consider and that soreness may guide the choice of exercises and intensity while incorrect exercise performance may be prevented by proper instruction. This may especially be of importance for the beyond working age version. DOMS may not be a problem for this type of exercise, but subjects should be prepared for transient general soreness if not accustomed to training. This is why we recommend an instructor to supervising the training – especially initially until participants are familiar with training. However, a detailed discussion on mechanisms seems beyond the scope of this paper that deals with the practical approach.
In section 2.1, the IPET program is mentioned, where are the characteristics of this concept defined?
Author reply: A reference is given already in paragraph 2 of the introduction. Further, we have now added a paragraph in the beginning of section 2 clarifying this concept more properly.
Figure 2.3 indicates the conceptual model of IPET, however, it does not indicate whether this model has been validated and what its objectives are in terms of the population that will be used.
Author reply: We have now in the beginning of the “section 2. The Concept of Intelligent Physical Exercise” added a paragraph including the most essential references e.g. also two review papers on IPET. In summary these review papers (the most recent from 2021) present the more than 20 RCT studies published - including more than 4000 workers - that have been conducted during the last two decades. The IPET concept has been developed over the years and a number of positive findings were evidenced, the main findings being: 1) reductions in musculoskeletal disorders, 2) reduced health risk indicators and improved physical capacities, and 3) increased productivity and monetary outcome on the company level.
Table 2.2 should be oriented horizontally and not vertically. If the authors look at this table it cannot be seen correctly.
Author reply: Table 2 is oriented horizontally - the layout is slightly changed to ease the reading.
The authors should indicate how they have table 2.2 in more detail.
Author reply: It is unclear to the authors what the referee means but we have revised the layout for better readability.
Authors should submit this topic and use the protocol of a systematic review to achieve an acceptable level of evidence. The following link provides information on this subject: http://www.equator-network.org/
Author reply: This is not a systematic review paper as specified above.
Performing an analysis of a topic without performing a search protocol of the selected articles makes the review have a biased view of the subject matter, this is because the authors' standard of the MDPI journals always indicates that they should receive systematic reviews, not bibliographic or narrative reviews.
Authors can also use the AMSTAR 2 protocol (Shea BJ, Reeves BC, Wells G, Thuku M, Hamel C, Moran J, et al. AMSTAR 2: a critical appraisal tool for systematic reviews that include randomised or non-randomised studies of healthcare interventions, or both. BMJ. 2017;358:j4008) if they consider PRISMA to be complex.
That said, authors should adapt all sections of this review or any of the suggested methodologies. To achieve that goal, this reviewer advises the authors to take a more specific and not so general approach. Perhaps it would be more interesting to focus on a specific type of population (e.g., noncommunicable diseases), as this would make it easier to demonstrate the efficacy of this program.
In the event that the authors do not agree to adapt this review to the PRISMA or AMSTAR 2 methodology, they can present to this reviewer according to the checklist advised by these guidelines how many items this review complies with and make the necessary changes so that this is reflected in the document. However, authors should be aware that this aspect should be included in a limitations section.
Author reply: The present paper is a state of knowledge with specific emphasis on a unique practical exercise prescription for the work-life population and beyond as mentioned above. The journal formalities did not allow to include a more established evidence-based methodology type of review for the demonstrated effectiveness of IPET. Based on this referee’s request we have now in the “section 2. The Concept of Intelligent Physical Exercise” added a paragraph including the most essential references e.g. also two review papers on IPET. The content of the reference list is unchanged, but the order has changes due to this paragraph, However, this change in order has not been marked. In summary these review papers (the most recent from 2021) present the more than 20 RCT studies published - including more than 4000 workers - that have been conducted during the last two decades. The IPET concept has been developed over two decades and a number of positive findings have been evidenced, the main findings being: 1) reductions in musculoskeletal disorders, 2) reduced health risk indicators and improved physical capacities, and 3) increased productivity and monetary outcome on the company level. For the convenience of this referee – and other readers not quite familiar with this research area – we have now added this information in a paragraph in the beginning of section 2. The present paper adds to the previous reviews by explaining the working algorithms that have been developed for the practical application of the IPET concept. One of the unique features is that it has a focus on the workplace and can be individualize to the broad presentation of non-communicable diseases present in the working age population.
The section on conclusions and future directions is usually done separately, not as a whole. However, according to the information that this review has been able to demonstrate, this section is too long and too broad.
Author reply: We find it relevant and important to combine these issues in the concluding section as it fits nicely with the “beyond” perspective in the title. As the other referees have no objection to this section, we prefer to keep the section unchanged.
References
Thirty percent of the literature is more than 10 years old and within this percentage are examples prior to the year 2000.
There are exceptions in which it is necessary to use older bibliography, however, this must be justified by the authors, since it is assumed that the articles presented should be those of greater novelty and relevance found within the subject under study.
Author reply: It is of high importance to acknowledge those studies that present breakthroughs in a research field. Standing on the shoulders of giants allows us to look further out. Repeated research reconfirming previous findings do not present the same novelty even if published within the latest 10 years. Therefore, purposely – knowing these giants – we have included some of their unique papers from the previous century. Additionally, a few method papers are included from before 2000, because they are still in general use and no better methods for those specific measures have been developed.
Reviewer 3 Report
Dear corresponding Author,
I appreciated your paper. It is interesting and it describes properly the issue.
I don't have any modification to suggest except to review the format and to adjust properly the tables beacuse in the present form it is very hard to read them correctly.
Author Response
Comments and Suggestions for Authors
Dear corresponding Author,
I appreciated your paper. It is interesting and it describes properly the issue.
I don't have any modification to suggest except to review the format and to adjust properly the tables beacuse in the present form it is very hard to read them correctly.
Author reply: We thank the referee for these encouraging words and are pleased that you find it interesting. We have checked the format to be consistent throughout the manuscript and adjusted Table 2.2. for better readability.
Round 2
Reviewer 2 Report
Dear Authors,
I appreciate your reply, however, it seems that you are not understanding the indications that are being provided to you. The review articles that have an acceptable level of evidence are the ones where the protocol has been validated, such as the ones I described above:
- The guideline advised by the author guidelines of this journal.
- The AMSTAR 2 protocol (Shea BJ, Reeves BC, Wells G, Thuku M, Hamel C, Moran J, et al. AMSTAR 2: a critical appraisal tool for systematic reviews that include randomised or non-randomised studies of healthcare interventions, or both. BMJ. 2017;358:j4008)
- The Scoping review protocol (Tricco AC, Lillie E, Zarin W, O'Brien K, Colquhoun H7, Kastner M, et al. A scoping review on the conduct and reporting of scoping reviews. BMC Med Res Methodol. 2016;16:15.)
- Stroup DF, Berlin JA, Morton SC, Olkin I, Williamson GD, Rennie D, et al. Meta-analysis of observational studies in epidemiology: a proposal for reporting. Meta-analysis Of Observational Studies in Epidemiology (MOOSE) group. JAMA. 2000;283(15):2008-12.
However, this reviewer has no problem in considering this narrative review as valid as long as before the conclusions section it should be indicated that the level of evidence of this review is limited and in the future any of the options given above should be carried out. In addition, the abstract should state next to the objective of this review that this article is a narrative or descriptive review that pursues the objective described in this article.
If the authors look at the references that justify the term IPET (6, 23, 25, 27 and 28), none is an evidence-based medical review according to the protocols described above. That said, even though such information has been published, this is a line of research that it is recommended to explore.
This is a new topic as can be seen in the searches carried out in PubMed:
Search: "Intelligent Physical Exercise Training"[Title/Abstract] OR IPET[Title/Abstract].
- Article identified without filter: 153.
- Article identified with meta-analysis and systematic review filter: no articles of interest.
Search: ("Intelligent Physical Exercise Training"[Title/Abstract] OR IPET[Title/Abstract]) AND (exercise [Title/Abstract]) Filters: Review
- Article identified without filter: 11.
- Article identified as a theoretical review: 2.
In Table 2.2 the authors indicate that the cut-off points are from the Danish population, two options are recommended to add other cut-off points from other populations/countries or before the conclusions section it should be indicated that in the future reviews should be carried out that include cut-off points from other populations. This information should also appear in the abstract.
Due to the recommendations indicated and the length of the conclusions, it is recommended that the authors synthesize this section. Although the authors have done a great job, they have not followed a validated protocol to select the information presented and therefore the evidence in the article has a limited level of evidence.
Best regards
Author Response
Reviewers comments
I appreciate your reply, however, it seems that you are not understanding the indications that are being provided to you. The review articles that have an acceptable level of evidence are the ones where the protocol has been validated, such as the ones I described above:
- The guideline advised by the author guidelines of this journal.
- The AMSTAR 2 protocol (Shea BJ, Reeves BC, Wells G, Thuku M, Hamel C, Moran J, et al. AMSTAR 2: a critical appraisal tool for systematic reviews that include randomised or non-randomised studies of healthcare interventions, or both. BMJ. 2017;358:j4008)
- The Scoping review protocol (Tricco AC, Lillie E, Zarin W, O'Brien K, Colquhoun H7, Kastner M, et al. A scoping review on the conduct and reporting of scoping reviews. BMC Med Res Methodol. 2016;16:15.)
- Stroup DF, Berlin JA, Morton SC, Olkin I, Williamson GD, Rennie D, et al. Meta-analysis of observational studies in epidemiology: a proposal for reporting. Meta-analysis Of Observational Studies in Epidemiology (MOOSE) group. JAMA. 2000;283(15):2008-12.
Author Reply: Of course, the authors of this paper have a fundamental knowledge about review articles, and we have tried to explain this reviewer that this paper is NOT a systematic review but a state of knowledge. We wrote in our last reply: “The present paper is a state of knowledge with specific emphasis on a unique practical exercise prescription for the work-life population and beyond ...” For this reason, we naturally still cannot follow the again above mentioned guidelines.
However, this reviewer has no problem in considering this narrative review as valid as long as before the conclusions section it should be indicated that the level of evidence of this review is limited and in the future any of the options given above should be carried out. In addition, the abstract should state next to the objective of this review that this article is a narrative or descriptive review that pursues the objective described in this article.
Author Reply: In the abstract we do not state this paper to be a review. We write: “The aim of this paper is to present an evidence-based exercise prescription for the work-life population and beyond.” The evidence is based on more than 20 RCT studies published - including more than 4000 workers - that have been conducted during the last two decades. Therefore, we do not find it relevant in the abstract to introduce the term of a review in any form.
If the authors look at the references that justify the term IPET (6, 23, 25, 27 and 28), none is an evidence-based medical review according to the protocols described above. That said, even though such information has been published, this is a line of research that it is recommended to explore.
Author Reply: The evidence we build our exercise prescription on are the more than 20 randomized controlled studies published. We disagree that only medical review papers can allow for the term “”evidence-based”. Evidence has certainly been published which this paper is based upon.
This is a new topic as can be seen in the searches carried out in PubMed:
Search: "Intelligent Physical Exercise Training"[Title/Abstract] OR IPET[Title/Abstract].
- Article identified without filter: 153.
- Article identified with meta-analysis and systematic review filter: no articles of interest.
Search: ("Intelligent Physical Exercise Training"[Title/Abstract] OR IPET[Title/Abstract]) AND (exercise [Title/Abstract]) Filters: Review
- Article identified without filter: 11.
- Article identified as a theoretical review: 2.
Author Reply: We agree with the reviewer, this is a new topic. This article is indeed exploring the concept IPET. However, as the IPET concept is new and not yet widely known, the literature search for the theoretical parts of this article was performed with other terms for intelligent physical exercise training than IPET or "Intelligent Physical Exercise Training", in order to broaden the search to well-known concepts. This is illustrated by an example from the literature search in Medline, where the concept is derived from the combination of the Population and Intervention search blocks as specified below:
(worksite or work site or workplace or work-place or work site or workforce or work-related or work environment or employee* or wage-earner* or occupation* or industrial).ti,ab,kw. or Occupations/ or exp Industry/ or exp occupational groups/ or Workforce/ or Workplace/
AND
((physical or cardio or aerobic or endurance or interval or high-intensity or resistance or strength or weight or functional or core or muscle or stretching or fitness) adj3 (training or exercise or activit*)).ti,ab,kw. or exp Exercise/ or muscle stretching exercises/ or resistance training/ or Endurance Training/ or High-Intensity Interval Training/
As shown, the literature search is not limited to the search terms of IPET or "Intelligent Physical Exercise Training"
In Table 2.2 the authors indicate that the cut-off points are from the Danish population, two options are recommended to add other cut-off points from other populations/countries or before the conclusions section it should be indicated that in the future reviews should be carried out that include cut-off points from other populations. This information should also appear in the abstract.
Author Reply: We have now in the legend for table 2.2. and in the “2.2. section” and “4.4. section summary” added that cut-off points from other populations are to be included whenever available.
Due to the recommendations indicated and the length of the conclusions, it is recommended that the authors synthesize this section. Although the authors have done a great job, they have not followed a validated protocol to select the information presented and therefore the evidence in the article has a limited level of evidence.
Author Reply: We have now in the “4.4. section summary” added that there is still limited evidence and further research as well as systematic reviews are requested.